Transcriptomic profiling of mTOR and ryanodine receptor signaling molecules in developing zebrafish in the absence and presence of PCB 95

Frank Daniel F. 1 2
http://orcid.org/0000-0002-1702-5592 Miller Galen W. 3
Connon Richard E. 1
http://orcid.org/0000-0001-7698-3443 Geist Juergen 2
Lein Pamela J. 3 pjlein@ucdavis.edu
1 Department of Anatomy, Physiology and Cell Biology, University of California , Davis, CA , USA
2 Department of Ecology and Ecosystem Management, Technical University Munich , Freising , Germany
3 Department of Molecular Biosciences, University of California , Davis, CA , USA
Heath Joan
Electronic publication date: 2017 Nov 29
Publication date: 2017
Volume: 5
Electronic Location ID: e4106
Received 2017 May 19; Accepted 2017 Nov 8
Copyright: © 2017 Frank et al.
Copyright year: 2017
Copyright holder: Frank et al.
License: This is an open access article distributed under the terms of the Creative Commons Attribution License, which permits unrestricted use, distribution, reproduction and adaptation in any medium and for any purpose provided that it is properly attributed. For attribution, the original author(s), title, publication source (PeerJ) and either DOI or URL of the article must be cited.
License URL: https://creativecommons.org/licenses/by/4.0/

Keywords: Transcriptomic profiling, mTOR, Persistent organic pollutants, Ryanodine receptor, Zebrafish, Neurodevelopmental disorders

Funding: National Institute of Environmental Health Sciences R01 ES014901 and F32 ES024070 United States Environmental Protection Agency RD 835550 Bayerische Forschungsstiftung DOK-169-14 This project was supported by the National Institute of Environmental Health Sciences (R01 ES014901 to Pamela J. Lein and F32 ES024070 to Galen W. Miller) and the United States Environmental Protection Agency (RD 835550 to Pamela J. Lein). The Bayerische Forschungsstiftung provided a postgraduate scholarship to Daniel F. Frank (contract no. DOK-169-14 to Juergen Geist). The funders had no role in study design, data collection and analysis, decision to publish, or preparation of the manuscript.

==============================
The mechanistic target of rapamycin (mTOR) and ryanodine receptor (RyR) signaling pathways regulate fundamental processes of neurodevelopment, and genetic mutations within these pathways have been linked to neurodevelopmental disorders. While previous studies have established that these signaling molecules are expressed in developing zebrafish, a detailed characterization of the ontogenetic profile of these signaling molecules is lacking. Thus, we evaluated the spatiotemporal expression of key transcripts in mTOR and RyR signaling pathways in wildtype zebrafish at 24, 72 and 120 hours post fertilization (hpf). We further determined whether transcriptional profiles of a subset of genes in both pathways were altered by exposure to PCB 95 (2,2′,3,5′,6-pentachlorobiphenyl), a pervasive environmental contaminant known to cause developmental neurotoxicity in mammalian systems via RyR-dependent mechanisms. Quantitative PCR revealed that transcription generally increased across development. Genes in the signaling pathway upstream of the mTORC1 complex, and the RyR-paralogs, ryr2a and ryr3, were robustly upregulated, and in situ hybridization of ryr3 coincided with a transcriptional shift from muscle to neuronal tissue after 24 hpf. Static waterborne exposure to PCB 95 beginning at 6 hpf significantly altered transcription of genes in both pathways. These changes were concentration- and time-dependent, and included downregulation of rptor, a member of the mTORC1 complex, at both 72 and 120 hpf, and increased transcript levels of the RyR paralog ryr2b and downstream target of RyR signaling, Wingless-type 2ba (wnt2ba) at 72 hpf. The detailed transcriptomic profiling of key genes within these two signaling pathways provides a baseline for identifying other environmental factors that modify normal spatiotemporal expression patterns of mTOR and RyR signaling pathways in the developing zebrafish, as illustrated here for PCB 95.

Introduction

Normal development of the nervous system requires the concomitant and coordinated ontogeny of specific signaling mechanisms in a temporally dependent and regionally dependent manner, and perturbations of either the temporal or quantitative aspects of any of these signaling events have been associated with altered patterns of neuronal connectivity, which are thought to underlie many neurodevelopmental disorders (NDDs) (Stamou et al., 2013). There is currently a significant interest in identifying chemicals that interact with signaling pathways implicated in the pathogenesis of NDDs in order to identify potential environmental risk factors for NDDs (Lyall et al., 2016b; Stamou et al., 2013). Larval zebrafish (Danio rerio) may be a particularly useful model for this purpose since zebrafish express homologs for >70% of human genes (Howe et al., 2013), and both the major stages of neurodevelopment and the signaling molecules that regulate neurodevelopment are highly conserved between zebrafish and humans (Gilbert, 2010). Moreover, larval zebrafish have proven to be a powerful model system for screening chemicals for potential neurotoxicity and therapeutic efficacy (Brady, Rennekamp & Peterson, 2016; Garcia, Noyes & Tanguay, 2016; MacRae & Peterson, 2015).

Two intracellular signaling pathways known to be critically important in the development of the nervous system are the mechanistic target of rapamycin (mTOR)-dependent and ryanodine receptor (RyR)-regulated signaling systems (Costa-Mattioli & Monteggia, 2013; Pessah, Cherednichenko & Lein, 2010). mTOR, a serine–threonine kinase, is conserved throughout the eukaryotic kingdom, and mTOR-dependent signaling pathways are critically important for integrating and controlling diverse cellular functions throughout life (Sarbassov, Ali & Sabatini, 2005; Wullschleger, Loewith & Hall, 2006). In the developing nervous system, mTOR signaling regulates early neurodevelopmental processes of cell growth and proliferation, as well as later stages of neurodevelopment, such as dendritic outgrowth and synaptogenesis (Kumar et al., 2005; Lee, Huang & Hsu, 2011). RyRs, which are expressed in all eukaryotes (Mackrill, 2012), represent a family of calcium-induced calcium release channels located in the endoplasmic reticulum (ER) of neuronal cells, where they function to regulate the release of calcium from internal stores (Pessah, Cherednichenko & Lein, 2010). RyR activity is important in calcium-dependent signaling pathways that regulate neuronal development and function (Pessah, Cherednichenko & Lein, 2010). Genetic mutations in either mTOR (Costa-Mattioli & Monteggia, 2013; Wang & Doering, 2013) or RyR (Matsuo et al., 2009; Soueid et al., 2016; Stamou et al., 2013) signaling pathways have been linked to NDDs.

Polychlorinated biphenyls (PCBs) are ubiquitous environmental pollutants that pose a significant risk to human health. Despite being banned from production in the late 1970s, environmental levels have not decreased significantly over the past decade (Martinez & Hornbuckle, 2011; Martinez et al., 2012). A primary endpoint of concern for human exposure to PCBs is developmental neurotoxicity. Multiple epidemiological studies have demonstrated an association between PCB exposures in utero or during infancy and neurological deficits in children (Berghuis et al., 2014, 2015; Korrick & Sagiv, 2008; Sagiv et al., 2012; Schantz, Widholm & Rice, 2003; Winneke, 2011), and more recently, PCBs have been identified as possible risk factors for NDDs, such as autism spectrum disorder, attention deficit and hyperactivity disorder and intellectual disability (Caspersen et al., 2016; Eubig, Aguiar & Schantz, 2010; Lyall et al., 2016a; Neugebauer et al., 2015; Nowack et al., 2015; Sealey et al., 2016). Non-dioxin-like PCB congeners are in particular associated with developmental neurotoxicity (Pessah, Cherednichenko & Lein, 2010; Stamou et al., 2013), and of these, PCB 95 (2,2′,3,5′,6-pentachlorobiphenyl) has been shown to disrupt normal patterns of neuronal connectivity in mammalian systems via RyR (Pessah, Cherednichenko & Lein, 2010; Wayman et al., 2012a, 2012b; Yang et al., 2009) and mTOR-dependent mechanisms (Miller and Lein, personal communication).

The goal of this study was to characterize the ontogenetic expression of key genes within the mTOR signaling pathway (Fig. S1), as well as RyR paralogs and selected genes involved in the regulation of RyR activity (Fig. S2), such as the RyR inhibitor homer1b (Feng et al., 2008), and genes whose expression is regulated by RyR activity, such as Wingless-type 2 paralogs, wnt2ba and wnt2bb (Wayman et al., 2012b). We also determined transcription of ER transmembrane proteins presenilin 1 and 2 (psen1, psen2), which influence the probability and frequency of RyR opening, and triadin (trdn), which forms a complex with RyRs to modulate channel opening depending on luminal Ca2+ status (Györke et al., 2004; Payne, Kaja & Koulen, 2015). Finally, we investigated transcription of two types of voltage dependent L-type calcium channels (cacna1c, cacna1sa), which regulate entry of extracellular calcium, thereby activating RyR to trigger calcium release from internal stores (Lipscombe, Allen & Toro, 2013). Expression of these genes was examined at 24, 72 and 120 hours post fertilization (hpf), which correspond to periods of early and late neurodevelopment in the zebrafish nervous system. In addition, we selected a subset of genes in both pathways (identified in bold font in Tables S1 and S2) to examine transcription following exposure of developing zebrafish to varying concentrations of PCB 95.

Materials and Methods

Chemicals

2,2′,3,5′,6-Pentachlorobiphenyl (PCB 95; 99.7% purity) was purchased from AccuStandard (New Haven, CT, USA).

Fish husbandry and spawning

Fish husbandry, spawning and all research involving zebrafish were performed in accordance with UC Davis Institutional Animal Care and Use Committee protocol #17645. Adult wild-type, tropical 5D zebrafish (D. rerio) were kept in 2 L tanks at a density of 10–14 fish at 28.5 ± 0.5 °C under a 14 h light:10 h dark cycle. Culture water pH was kept in the range of 7.2–7.8, and electric conductivity between 600 and 800 μS cm−1. Adult fish were fed twice a day with Artemia nauplii (INVE Aquaculture, Inc., Salt Lake City, UT, USA) and commercial flake (a combination of Zebrafish Select Diet, Aquaneering, San Diego, CA, USA and Golden Pearls, Artemia International LLC, Fairview, TX, USA). Embryos were obtained by spawning groups of 8–12 fish in a 1:2 female/male ratio. Spawning time was coordinated using a barrier to separate male and female fish, which was removed in the morning after the lights turned on, thus producing age-matched fertilized eggs, which were collected within 20 min of spawning.

Quantitative polymerase chain reaction

For the initial studies of the normal ontogenetic profiles of transcription of mTOR and RyR signaling molecules, embryos were directly transferred into 100 × 20 mm polystyrene tissue culture dishes (Corning Inc., Corning, NY, USA) containing 60 mL standardized Embryo Medium (Westerfield, 2007) and placed into an incubator at a constant temperature of 28.5 ± 1 °C, with a 14 h light:10 h dark photoperiod. Fish were maintained at a density of 50 individuals per petri dish, and RNA was extracted at three different time points (24, 72 and 120 hpf). Fish from three independent spawns were used to obtain three biological replicates at each time point. To minimize variability due to differences in spawning time, all embryos were collected within a 20 min spawning window. Fish within each spawn were pooled into batches of 350 or 500 to extract sufficient amounts of RNA for quantitative polymerase chain reaction (qPCR) analyses. The first biological replicate represented a pooled sample of 500 embryos in order to collect a sufficient amount of RNA for method validation. Embryo numbers were chosen using a conservative approach of assessing average population transcription of target genes, rather than gene expression from single individuals. We quantified baseline transcription of 36 genes, 23 associated with the mTOR signaling pathway (Table S1) and 13 associated with RyR-dependent Ca2+ signaling (Table S2).

Transcript levels of target genes were assessed by qPCR using gene-specific primers derived from mRNA sequences obtained from the Zebrafish Model Organism Database (http://zfin.org/). Primers were designed using NCBI Primer Blast (http://www.ncbi.nlm.nih.gov/tools/primer-blast/), and obtained from Integrated DNA Technology (Integrated DNA Technologies, Inc., Coralville, IA, USA).

RNA extractions were performed with a Qiagen Qiacube robotic workstation using RNeasy Mini Kit spin columns (Qiagen, Valencia, CA, USA) as per the manufacturer’s directions. Total RNA integrity was verified using an RNA 6000 Nano Kit on an Agilent Bioanalyzer 2100 (Agilent Technologies, Santa Clara, CA) (Fig. S3). The RNA Integrity Ratio (RIN) scores were calculated, and RNA was deemed to be of good quality with RIN scores of over 8 (Table S3). RNA concentration was determined using a NanoDrop ND1000 spectrophotometer (NanoDrop Technologies, Inc., Wilmington, DE, USA).

Complementary DNA (cDNA) was synthesized using 1 μg total RNA in a reaction with 4 μL Superscript Vilo Mastermix (SuperScript® VILO™ MasterMix, Invitrogen, Carlsbad, CA, USA) according to the user’s manual. Reactions were incubated for 10 min at 25 °C, 60 min at 42 °C followed by a 5 min denaturation step at 85 °C. Samples were then diluted with nuclease-free water in a 1:5 ratio to produce acceptable concentrations for quantitative PCR evaluations. Success of the cDNA-synthesis was tested using beta-actin primers with a polymerase chain reaction (5 min at 95 °C; 30 s at 95 °C, 30 s at 60 °C, 45 s at 72 °C, in 35 cycles; 10 min at 72 °C) visualized through gel electrophoresis.

Quantitative polymerase chain reaction was conducted using Power SYBR Green PCR Master Mix (Life-technologies, Carlsbad, CA, USA). Primer validation was performed using a seven point standard curve with three replicates; amplification efficiencies ranged between 90.1% and 108.7%. Cycling conditions were 2 min at 50 °C, 10 min at 95 °C, 40 cycles of 15 s at 95 °C, 30 s at 60 °C and 30 s at 72 °C, followed by a thermal ramping stage for dissociation evaluation. Amplification data were analyzed using Sequence Detection Systems software (SDS v2.4.1, Applied Biosciences). Relative gene expression was calculated using the Log2−ΔΔCT method (Livak & Schmittgen, 2001) relative to the reference genes elongation factor 1 alpha (eef1a1) and beta actin 2 (actb2) (Table S2), which sustained best scores in GeNorme (Vandesompele et al., 2002). All data were normalized to 24 hpf samples. To verify primer quality, sequences obtained from NCBI were checked additionally in Ensembl genome browser (http://www.ensembl.org/index.html) to identify chromosome and exon location (Tables S1 and S2).

Whole mount in situ hybridization

Digoxigenin-labeled probes were prepared from 24 hpf embryonic cDNA for ryr3. We chose ryr3 because it experiences a highly significant increase in transcription between 24 and 72 hpf as well as between 24 and 120 hpf, and because of discrepancies in the published data, which describe ryr3 transcription in zebrafish prior to 24 hpf in skeletal muscle only, or predominantly in hindbrain regions at later time points (Thisse & Thisse, 2005; Wu, Brennan & Ashworth, 2011). Specifically, we designed primers (Table S2) containing a T3 RNA polymerase promoter on the 5′-end of the reverse primer, thereby allowing antisense probe transcription. Procedures for embryo and larval preparation and in situ hybridization assay were performed as described previously (Thisse & Thisse, 2014), using BM purple (Roche, Basel, Switzerland) as labeling solution.

PCB 95 exposures

Embryos collected from spawning tanks were transferred to 100 × 20 mm polystyrene tissue culture dishes (Corning Inc., Corning, NY, USA) and maintained at a constant temperature of 28.5 ± 1 °C. At 4 hpf, embryos were enzymatically dechorionated using 50 μL of 41 mg/mL pronase (Sigma-Aldrich, St. Louis, MO, USA) in 25 mL of culture water for a maximum of 6 min (Truong, Harper & Tanguay, 2011). Embryos were then allowed to recover for 2 h in culture water before being transferred one embryo per well into a 96-well plate (Falcon™; Corning Inc., Corning, NY, USA) containing 100 μL of standardized Embryo Medium (Westerfield, 2007). At 6 hpf, 100 μL of 2× PCB 95 solution was added directly into each well to yield final concentrations of 0.1, 0.3, 1.0, 3.0 or 10.0 μM PCB 95; control embryos were exposed to vehicle (0.2% DMSO). Plates were covered with Parafilm M (Bemis NA, Neenah, WI, USA) to reduce evaporation, and were then placed into an incubator at a constant temperature of 28.5 ± 1 °C and a 14 h light:10 h dark cycle until fish (n = 16 per treatment in each plate) were harvested for transcriptomic assessments at 72 and 120 hpf. Embryos or larvae were pooled into batches of 12–16 individuals to generate sufficient RNA for qPCR analyses. Fish from three independent spawns were used to obtain three biological replicates at each time point.

Statistical analysis

Significant differences in gene transcription at 72 and 120 hpf relative to transcription at 24 hpf were identified using one-way ANOVA with significance set at P < 0.05 followed by a Tukey’s post hoc test. If data did not fit the ANOVA assumptions of normality, a Kruskal–Wallace test was applied (P < 0.05), followed by the Nemenyi–Damico–Wolfe–Dunn post hoc test. Shapiro–Wilk normality and Bartlett tests were used to determine which algorithms are appropriate for determining significant differences between time points. R-packages “stats” (Team, 2014), “PMCMR”; pairwise multiple comparisons of mean rank sums (Pohlert, 2014) and “multcomp”; multiple components (Hothorn et al., 2008) were used to perform statistical analyses. Data from the PCB 95 exposure studies were similarly analyzed, with the different PCB exposures normalized to the solvent control.

Results

A critical first step in developing a zebrafish platform to screen for gene × environment interactions of relevance to NDDs is to establish the normal ontogenetic profile of NDD-relevant signaling molecules in the developing zebrafish. Therefore, we first characterized the ontogenetic expression profiles of 36 genes, 23 associated with the mTOR signaling pathway (Table S1) and 13 associated with RyR-dependent Ca2+ signaling (Table S2).

Ontogenetic profile of transcripts encoding mTOR signaling pathway genes

Overall, there was increased transcription of genes encoding signaling molecules upstream of mTOR between 24 and 120 hpf (Figs. 1A and 1B). Ribosomal protein S6 kinase polypeptide 1 (rps6ka1) was significantly upregulated (P < 0.05) at both 72 and 120 hpf relative to 24 hpf, while tuberous sclerosis 1a (tsc1a) and mitogen-activated protein kinase 1 (mapk1) were significantly upregulated (P < 0.05) at 120 hpf relative to 24 hpf. Mitogen-activated protein kinase 3 (mapk3) and insulin receptor substrate 1 (irs1) remained constant across all time points, and V-akt murine thymoma viral oncogene homolog 1 (akt1) showed a declining trend in relative expression over time. All other upstream targets of the mTOR complex exhibited a general increased transcription from 24 to 120 hpf that was not statistically significant.

Figure 1 Relative expression of transcripts encoding mTOR signaling molecules.

(A, B) Signaling molecules upstream of mTOR; (C) mTOR Complex 1 and 2; (D–F) signaling molecules downstream of mTOR. Data presented as the mean ± SE (n = 3 independent biological replicates). Within each sample, values for the target transcript were normalized to the average of the values for the reference genes, actb2 and eef1a1, within that same sample. *Significantly different from 24 hpf at P < 0.05; † significantly different from 72 hpf at P < 0.05 as determined by one-way ANOVA followed by Tukey’s post hoc test, or if data did not meet the ANOVA assumptions, as determined by the Kruskal–Wallace test followed by a Nemenyi–Damico–Wolfe–Dunn post hoc test.

There were no significant differences in transcription of genes within the mTORC1 complex from 24 to 72 or 120 hpf (Fig. 1C). However, relative to transcript levels at 24 hpf, transcription of raptor-independent companion of mTOR (rictora) decreased at 72 hpf but was significantly upregulated from 72 to 120 hpf. Transcription of genes encoding signaling molecules downstream of mTOR generally decreased with increasing hpf, but none of these changes were statistically significant (Figs. 1D–1F). The one exception was the p70 ribosomal S6 kinase a (rps6kb1a), which showed the tendency of increased transcription at 72 hpf, although this change was not statistically significant.

Baseline transcription of genes involved in RyR-dependent signaling

Zebrafish express five RyR-paralogs (Wu, Brennan & Ashworth, 2011): ryr1a and ryr1b are transcribed predominantly in slow and fast twitch muscle tissue, respectively (Hirata et al., 2007); ryr2a is predominantly expressed in the central nervous system (CNS), ryr2b, in the heart of developing zebrafish (Wu, Brennan & Ashworth, 2011); and ryr3 has been detected in zebrafish skeletal muscle prior to 24 hpf (Wu, Brennan & Ashworth, 2011) but is reported to be expressed predominantly in hindbrain regions at later developmental stages (Thisse & Thisse, 2005). Transcripts of ryr2a and ryr3 were significantly elevated at 72 and 120 hpf, whereas relative expression of ryr1a, ryr1b and ryr2b did not change significantly over the 120 hpf assessment period (Fig. 2A). Transcription of wnt2ba increased in a time-dependent manner with a significant increase noted at 120 hpf, whereas wnt2bb did not change significantly over the 120 hpf assessment (Fig. 2B).

Figure 2 Relative expression of transcripts encoding ryanodine receptor (RyR) paralogs and regulatory molecules at 24, 72 and 120 hpf.

(A) Zebrafish RyR paralogs; (B) Wnt2 orthologs; (C, D) RyR regulatory molecules. Data are shown as the mean ± SE (n = 3 independent biological replicates). Within each sample, values for the target transcript were normalized to the average of the values for the reference genes, actb2 and eef1a1, within that same sample. *Significantly different from 24 hpf at P < 0.05, **P < 0.01, ***P < 0.001, as determined by one-way ANOVA followed by Tukey’s post hoc test, or if data did not meet the ANOVA assumptions, as determined by the Kruskal–Wallace test followed by the Nemenyi–Damico–Wolfe–Dunn post hoc test.

Significant increases in gene transcription were observed for Homer homolog 1b (homer 1b) and the alpha 1c subunit of the voltage-dependent L type calcium channel (cacna1c) at 72 and 120 hpf. Transcript levels of the alpha 1S subunit 1 of the voltage-dependent L type calcium channel (cacna1sa) were significantly increased at 72 hpf (Fig. 2C). Transcription of psen1 and psen2 was elevated at 72 hpf, but this change was not significant (Fig. 2C), whereas genes involved in regulating RyR-dependent signaling (like trdn) remained relatively consistent across the three time points evaluated in this study.

In-situ hybridization of ryr3

Since ryr3 is expressed in both the skeletal muscle and brain at different developmental stages (Thisse & Thisse, 2005; Wu, Brennan & Ashworth, 2011), to determine whether increased transcripts of ryr3 detected by qPCR reflect ryr3 upregulation in the CNS, we examined its expression by in situ hybridization. At 24 hpf, ryr3 transcription was observed only in skeletal muscle (Fig. 3A). However, by 26 hpf, ryr3 transcripts were detected in brain tissue (Figs. 3B and 3C). By 72 hpf, ryr3 was abundantly expressed in the brain (Fig. 3D).

Figure 3 Spatial expression patterns of ryr3 transcripts as determined by in situ hybridization.

To obtain dorsal views, the yolk sac was removed prior to imaging. (A) Dorsal view of ryr3 expression at 24 hpf; expression is predominantly in fast twitch muscles and somites. (B) Dorsal view of ryr3 expression at 26 hpf; in addition to expression in fast twitch muscles and somites; ryr3 mRNA is expressed in the hindbrain and telencephalon. (C) Lateral view of ryr3 expression at 26 hpf. Transcripts for ryr3 are present in the whole organism with higher intensity in the telencephalon and habenula. (D) Dorsolateral view of ryr3 expression at 72 hpf, to specially highlight expression in the brain. Transcripts for ryr3 are detected in fast twitch muscles and somites, but are more abundantly expressed in the hindbrain and telencephalon. The dotted line highlights the mid-line of the brain (from a dorsal view).

Transcriptional effects of PCB 95

The influence of developmental PCB 95 exposure on transcription of NDD-relevant genes was tested on six genes associated with the mTOR signaling pathway (Table S1; genes in bold font) and six associated with RyR-dependent Ca2+ signaling (Table S2; gene in bold font) (Fig. 4). The mTORC1 member, rptor, was significantly downregulated in PCB 95-exposed fish at both 72 and 120 hpf (Fig. 4B). At both time points, the effect of PCB 95 was concentration-dependent. At 72 hpf, the fold-change progressively decreased with increasing concentrations of PCB 95, with statistical significance reached at the highest PCB 95 concentration of 10 μM. At 120 hpf, statistically significant downregulation was observed at 3.0 and 10.0 μM. The other five genes of the mTOR signaling pathway that we examined were not significantly altered at 72 or 120 hpf in fish exposed to PCB 95 at any of the concentrations tested in this study (Figs. 4A, 4C–4F). Within the subset of genes relevant to RyR signaling that we examined, two were found to be significantly upregulated by PCB 95 exposure but interestingly, only at 72 hpf (Figs. 4K and 4L). Transcription of ryr2b and wnt2ba showed a concentration-dependent increase, which reached statistical significance at 10 μM PCB 95.

Figure 4 Concentration-dependent effects of PCB 95 on transcription of mTOR and ryanodine receptor (RyR) signaling pathways at 72 and 120 hpf.

Fold-change in transcription of genes coding for (A–F) mTOR signaling molecules and (G–L) RyR paralogs and Wingless-type 2ba. The solid line depicts concentration-dependent effects of PCB 95 at 72 hpf; the dashed line, 120 hpf. Data are shown as the mean ± SE (n = 3 independent biological replicates). Within each sample, values for the target transcript were normalized to the average of the values for the reference genes, actb2 and eef1a1, within that same sample. Significant differences from vehicle controls at the same time point are identified by *P < 0.05, **P < 0.01, ***P < 0.001 as determined by one-way ANOVA followed by Tukey’s post hoc test, or if data did not meet the ANOVA assumptions, as determined by the Kruskal–Wallace test followed by the Nemenyi–Damico–Wolfe–Dunn post hoc test.

Discussion

This study provides the most comprehensive characterization to date of the ontogenetic profile in developing zebrafish of 36 transcripts encoding molecules involved in mTOR and RyR signaling pathways. All 23 genes examined in the mTOR signaling cascade were detected at the mRNA level at all three developmental time points. However, only four were observed to be differentially regulated during the first 120 hpf: rps6ka1, mapk1 and the mTORC2 complex member rictora, all of which encode molecules that promote mTOR signaling (Anjum & Blenis, 2008; Guertin et al., 2006), and tsc1a, which encodes the tumor sclerosis complex protein, hamartin that inhibits mTOR signaling. All four genes were significantly upregulated, but at differing developmental stages: rps6ka1 and mapk1 were upregulated at 72 hpf; rictora and tsc1a, at 120 hpf. However, transcript levels for the majority of mTOR-related genes were relatively stable throughout the first five days of development, suggesting that chemical perturbation of transcription of these genes may have significant adverse impacts.

Consistent with previous studies (Wu, Brennan & Ashworth, 2011), we observed that mRNA expression of the muscle-specific (ryr1a and ryr1b) and heart-specific (ryr2b) RyR homologs also remained largely unchanged throughout early zebrafish development. In contrast, transcript levels of ryr2a, the CNS-specific RyR paralog, and ryr3 were significantly elevated at 72 and 120 hpf, corresponding to the peak period of synaptogenesis (Brustein et al., 2003; Saint-Amant & Drapeau, 1998). As reported by others (Wu, Brennan & Ashworth, 2011), we did not detect ryr3 in the brain prior to 24 hpf as determined by in situ hybridization. However, brain expression of ryr3 was apparent in the whole mount as early as 26 hpf, and by 72 hpf, ryr3 was strongly expressed throughout the brain, and to a lesser extent in the whole body. These in situ hybridization data suggest that increased ryr3 mRNA detected by qPCR reflects upregulation of ryr3 in the CNS.

Mammals have three RyR paralogs: ryr1, ryr2, and ryr3 (Pessah, Cherednichenko & Lein, 2010), and dynamic changes in the spatiotemporal expression of all three paralogs have been described in the developing mouse brain (Mori et al., 2000). Unlike zebrafish, ryr1 and ryr2 transcription in developing rodents significantly increases in cardiac and muscle tissue during the first days after birth, and become more abundant with ongoing embryogenesis (Brillantes, Bezprozvannaya & Marks, 1994; Rosemblit et al., 1999). However, similar to zebrafish, developing mice exhibit reduced ryr3 expression in muscle and increased expression in the brain with ongoing development (Bertocchini et al., 1997; Mori et al., 2000; Takeshima et al., 1996).

Several genes that encode proteins important in the regulation of RyR activity were also found to be differentially expressed in developing zebrafish. Transcription of wnt2ba and cacna1c was upregulated at 72 and 120 hpf; whereas cacna1sa transcripts were significantly increased at 72 hpf but decreased at 120 hpf. Wnt2 transcription is linked to activity-dependent dendritic outgrowth in mammalian neurons (Wayman et al., 2006), cacna1c is described in rodents as the most abundant L-type Ca2+ channel in neurons, and cacna1sa is found in rodent muscle tissue. Mutations in all three genes have been linked to NDDs (Caracci, Vila & De Ferrari, 2016; Kim & State, 2014; Kwan, Unda & Singh, 2016; Trevarrow, Marks & Kimmel, 1990). We also observed a significant increase in homer1b transcripts, which is consistent with published data demonstrating homer1b transcription in the myotome and nervous system during early stages of development that shifts toward the brain around 48 hpf (Thisse & Thisse, 2004). Similarly, developing exhibit predominant neural transcription of homer1b with low transcript levels in other organs (Shiraishi-Yamaguchi & Furuichi, 2007).

This study also demonstrated that developmental exposure to PCB 95 significantly altered transcription of a subset of mTOR and RyR signaling molecules in larval zebrafish. PCB 95 is a developmental neurotoxicant that promotes activity-dependent dendritic growth in primary rat neurons via RyR-dependent upregulation of wnt2 transcription (Wayman et al., 2012a, 2012b) and activation of mTOR-dependent translational mechanisms (G. Miller, 2017, personal communication). PCB 95 caused a time- and concentration-dependent downregulation of rptor and upregulation of ryr2b and wnt2ba. These effects are not likely to be secondary consequences of decreased viability of fish exposed to PCB 95 because: (1) separate studies have observed no morbidity or teratogenic effects in zebrafish exposed to PCB 95 at 10 μM from 6 to 120 hpf (G. Miller, 2017, personal communication); and (2) the majority of genes (9 of 12) examined were not significantly altered by PCB 95 exposure. These findings are consistent with a recent report that ryr2 transcripts are upregulated in developing Atlantic killifish (Fundulus heteroclitus) from New Bedford Harbor that are exposed to high levels of non-dioxin-like PCBs (Fritsch et al., 2015). However, mtor transcripts were also found to be upregulated in the New Bedford Harbor killifish but not in our PCB 95-exposed zebrafish. This difference likely reflects species-dependent responses and/or differences in the exposures (complex environmental exposures vs. defined exposure to a single PCB congener).

The functional relevance of PCB 95 effects on transcription of mTOR and RyR signaling molecules in developing zebrafish remains to be determined. Experimental studies have shown that functional knockout of raptor significantly reduces dendritogenesis in the mammalian brain (Cloëtta et al., 2013; Urbanska et al., 2012), which is the opposite of PCB 95 effects on dendritic arborization. This suggests that PCB 95 downregulation of rptor is not causally related to PCB 95 effects on dendritic growth. Similarly, upregulation of ryr2b is likely not involved in the dendrite promoting activity of PCB 95 because the expression of this RyR paralog is limited to cardiac tissue (Wu, Brennan & Ashworth, 2011). However, the observation that PCB 95 upregulated expression of wnt2ba in developing zebrafish is consistent with observations of increased levels of wnt2 transcripts in rat hippocampal neurons exposed to PCB 95, an effect that was causally linked to PCB 95 effects on dendritic arborization (Wayman et al., 2012b).

In conclusion, this study provides fundamental data regarding the transcriptional profiles of major components of the mTOR and RyR signaling pathways during the first five days of zebrafish development. Most genes differentially regulated during development were upregulated at times corresponding to peak synaptogenesis and formation of neuronal circuits, consistent with evidence implicating a major role for both pathways in normal neurodevelopment, and in the pathogenesis of NDDs. Developmental exposure to PCB 95 altered transcription of wnt2ba, a gene implicated PCB 95 developmental neurotoxicity in mammalian models (Stamou et al., 2013; Wayman et al., 2012b), demonstrating the feasibility of using the zebrafish to screen for chemicals that modulate expression of mTOR and RyR signaling pathways to identify potential environmental risk factors and/or therapeutics of relevance to NDDs.

Supplemental Information

Supplemental Information 1 Supplemental material.

Click here for additional data file.

We gratefully acknowledge Paige Mundy (University of California, Davis, Connon laboratory), who assisted with generating data regarding quality of the RNA used for qPCR analyses.

Additional Information and Declarations

Competing Interests

Author Contributions

Animal Ethics

Data Availability

The authors declare that they have no competing interests.

Daniel F. Frank conceived and designed the experiments, performed the experiments, analyzed the data, wrote the paper, prepared figures and/or tables, reviewed drafts of the paper.

Galen W. Miller conceived and designed the experiments, analyzed the data, wrote the paper, reviewed drafts of the paper.

Richard E. Connon conceived and designed the experiments, analyzed the data, wrote the paper, reviewed drafts of the paper.

Juergen Geist analyzed the data, wrote the paper, reviewed drafts of the paper.

Pamela J. Lein conceived and designed the experiments, analyzed the data, contributed reagents/materials/analysis tools, wrote the paper, reviewed drafts of the paper.

The following information was supplied relating to ethical approvals (i.e., approving body and any reference numbers):

All research reported in this manuscript involving vertebrate animals was approved by the Animal Care and Use Committee of the University of California, Davis.

The following information was supplied regarding data availability:

The raw data has been provided as a Supplemental File.

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
