# Peer review of "Transcriptomic profiling of mTOR and ryanodine receptor signaling molecules in developing zebrafish in the absence and presence of PCB 95"

_PeerJ, doi:10.7717/peerj.4106_

## Round 0.1 · original submission · Major Revisions

Dear authors,

I agree with both reviewers that your study is of a descriptive nature and of very limited scope. For this article to be published in PeerJ, all the reviewers' suggestions need to be addressed very carefully. It would be ideal if you could now add data of an experimental nature to the findings, but if not, we would expect to receive a much shorter article with all the advice about consolidating figures, moving figures to Supplementary data and presenting a greatly reduced Discussion section taken on board.

Reviewer 1 ·

Basic reporting

Well-written paper, but some of the figures are not necessary or could be moved to supplemental information. Raw data were not shared as supplemental information. Discussion is too long given the limited scope of the study.

Experimental design

Solid study but very limited in scope, as the authors only evaluated baseline expression profiles of a handful of genes at three stages of embryonic/larval development.

Validity of the findings

Findings are valid, but the authors should just present relative expression or absolute transcript numbers (not both).

Additional comments

Line 11: Please spell out mTOR upon first use.

Lines 60-61: Recommend moving Figures 1 and 2 to supplemental information.

Lines 230-343: The Discussion is way too long (~5 pages), especially given the limited scope of this study. Therefore, the Discussion could easily be trimmed down to no more than 2.5-3 pages.

Lines 262-262: While this is true, it's unknown whether chemical uptake into the yolk sac interferes with nutrient availability and utilization during embryonic development. Please revise this sentence to address this possibility.

Tables 1 and 2: Move to supplemental information.

Figures 3 and 4: These two figures could easily be combined into a single heat-map or volcano plot.

Figure 5: I'm not convinced that these data add value in addition to Figures 3 and 4. Why not just present as a fold-change or absolute copy number? Both are not necessary, as these figures rely on the same qPCR data.

Figure 6: Please include images for sense ryr3 RNA probes (negative controls) in order to account for potential non-specific hybridization in situ.

Reviewer 2 ·

Basic reporting

The article is well written and the background for the study is sufficient. The figures and tables are professional. There are no hypotheses stated as this is an observational study of gene expression during ontogeny of zebrafish.

Experimental design

In general the experimental design section is well written. There is a question about how the RNA check gel was performed and it would be good to have included a picture for the quality of the RNA.

Validity of the findings

The findings are valid but the study is observational. There was no experiment included.

Additional comments

In this manuscript, the authors map gene expression of key transcripts in the mTOR and RyR signaling pathways in wild type zebrafish embryos at three times points during development. The genes that they selected are excellent genes for these pathways and they provide primer pairs for the genes of interest. While it is interesting to know the timing of expression of these genes, I found the study to be descriptive and without an experimental hypothesis. Apparently this study is setting the baseline for future studies in which they will examine changes in these genes after exposure to contaminants. In my opinion, the study would be improved if the authors incorporated an actual experiment.

Comments:
1. The authors extract the total RNA from a pooled set of whole embryos for a set of genes that might give information for brain development. Since these genes are likely expressed differentially in tissues during organogenesis, they dilute their signal by using whole embryos. This is a major limitation of the study as the authors point out in line 292. They might get better results if they could micro-dissect out the various tissues of interest.
2. How closely were the fish embryos that were pooled synchronized for development? Since development is rapid, a half hour difference for pooled embryos could erase a significant change in gene expression.
3. Line 207 – absolute transcript copy numbers -- I have an issue with this. The authors make cDNA for the amplified segment and purify it and measure the quantity by nanodrop. However, when doing PCR from a transcript, there is the step of reverse transcriptase that is not necessarily 100% efficient. If they want to have absolute copy number, they should start with RNA. They could get RNA for each of their genes by adding a T3 RNA polymerase promoter to one end and transcribing the RNA and quantifying that product before starting the PCR part. Also it is not clear if the absolute transcript copy number is per embryo or per ug total RNA or??? It would probably be most interesting to have the calculation be per embryo.
4. Line 43-45 – statement that the two pathways are critical for the development of the nervous system needs a reference. It seems to me that mTOR would be critical for the whole organism as would the RyR signaling pathways since they both play critical roles in homeostasis.
5. Line 114. Was total RNA integrity measured on a denaturing agarose gel – for example MOPS gel? It might be good to provide an example of the gel since RIN numbers are not provided.
6. Line 186 – in the text the gene is called “rictora” but in the figures it is referred to as “rictor” Please correct and make them the same.
7. Line 228. I don’t think that it is possible to quantitatively compare the staining for in situ across the different embryos shown in Figure 6. The magnification seems to be different for the embryos and there should be a standard against which to measure the intensity of the stain for quantitation. The statement …. “even though expression throughout the rest of the body is still low…” doesn’t seem to match the very dark staining seen in the rest of the body in the other panels at earlier stages without appearance in the brain.
8. Line 249 -- I don’t think the authors can comment on whether the increased expression of rictora and rptor promote mTOR activity since (a) they have not measured mTOR activity and (b) mTOR activity is promoted at the level of phosphorylation
9. Line 253 – Chemical perturbations may change the expression patterns for these genes, but the authors have not performed these experiments. If they have, they should include them in the paper as it would strengthen the paper.
10. Line 263 – not all yolk sacs are nutritionally equivalent and provide the same amount of nutrients. There are some eggs of lower quality that may have inferior yolk composition, especially if mom was exposed to contaminants.
11. Lines 307-310. The authors have not shown in situ for Ryr1 or Ryr2 and thus cannot comment on spatial expression in the brain in zebrafish in their experiment. They quote other authors, suggesting that RyR spatial expression is already known for zebrafish and thus takes away the novelty for this study.

Minor comments
Line 47 delete “the” should read …throughout lifespan….
Line 91 Add “was” should read….and RNA was extracted…..
Line 246 rictora or rictor?

---

## Round 0.2 · Minor Revisions

Thank you very much for the revised version of your manuscript which is a significant improvement on the original. The only question that remains from my perspective concerns the quality of the RNA that was used for the gene expression studies. This is a very important point. In your response to reviewers you mention checking the RNA quality and yield by nanodrop and by running the DNA on an agarose gel. Neither of these methods is appropriate. Firstly, a nanodrop cannot distinguish degraded RNA from intact RNA (it measures nucleotide absorbance whether the RNA is intact or not); secondly, RNA integrity should be checked on a denaturing agarose gel, rather than a normal DNA gel. Hopefully, you have some of the RNA left that was used in the experiments. If so, please run it out on a denaturing gel. If not, please provide the DNA gel for inspection.

Reviewer 1 ·

Basic reporting

The authors have adequately addressed all previous comments.

Experimental design

The authors have adequately addressed all previous comments.

Validity of the findings

The authors have adequately addressed all previous comments.

Additional comments

Recommend accepting for publication.

---

## Round 0.3 · accepted · Accept

The RNA integrity data is perfectly satisfactory, and improves the manuscript. However, the corresponding section in Methods is still written inaccurately so I have taken the liberty of correcting it myself on the submitted manuscript.

Would the corresponding author please verify that my corrected wording (see below) is acceptable to them?

RNA extractions were performed with a Qiagen Qiacube robotic workstation using RNeasy Mini Kit spin columns (Qiagen, Valencia, CA, USA) as per the manufacturer’s directions. Total RNA integrity was verified using an RNA 6000 Nano Kit on an Agilent Bioanalyzer 2100 (Agilent Technologies, Santa Clara, CA) (Supplemental Figure S3). The RNA Integrity Ratio (RIN) scores were calculated, and RNA was deemed to be of good quality with RIN scores of over 8 (Supplemental Table S3). RNA concentration was determined using a NanoDrop ND1000 spectrophotometer (NanoDrop Technologies, Inc., Wilmington, DE, USA).